# IMTLAB: An Open-Source Platform for Building, Evaluating, and Diagnosing Interactive Machine Translation Systems

**Xu Huang**[1*]    **Zhirui Zhang**[2]    **Ruize Gao**[3]    **Yichao Du**[4]    **Lemao Liu**[2]
**Gouping Huang**[2]    **Shuming Shi**[2]    **Jiajun Chen**[1]    **Shujian Huang**[1†]

[1]National Key Laboratory for Novel Software Technology, Nanjing University
[2]Tencent AI Lab    [3]Shanghai Jiao Tong University
[4]University of Science and Technology of China
[1]xuhuang@smail.nju.edu.cn, {chenjj,huangsj}@nju.edu.cn
[2] zrustc11@gmail.com, {redmondliu,donkeyhuang,shumingshi}@tencent.com
[3] ruizgaonlp@gmail.com    [4] duyichao@mail.ustc.edu.cn

## Abstract

We present IMTLAB, an open-source end-to-end interactive machine translation (IMT) system platform that enables researchers to quickly build IMT systems with state-of-the-art models, perform an end-to-end evaluation, and diagnose the weakness of systems. IMTLAB treats the whole interactive translation process as a task-oriented dialogue with a human-in-the-loop setting, in which human interventions can be explicitly incorporated to produce high-quality, error-free translations. To this end, a general communication interface is designed to support the flexible IMT architectures and user policies. Based on the proposed design, we construct a simulated and real interactive environment to achieve end-to-end evaluation and leverage the framework to systematically evaluate previous IMT systems. Our simulated and manual experiments show that the prefix-constrained decoding approach still gains the lowest editing cost in the end-to-end evaluation, while BiTI-IMT (Xiao et al., 2022) achieves comparable editing cost with a better interactive experience.

## 1 Introduction

In recent years, there has been significant development in neural machine translation (NMT) (Bahdanau et al., 2015; Vaswani et al., 2017; Hassan et al., 2018). However, the quality of machine-translated texts still cannot meet the rigorous demands of industrial applications, necessitating costly and inefficient human intervention. Interactive machine translation (IMT) (Foster et al., 1997; Langlais et al., 2000; Barrachina et al., 2009; Cheng et al., 2016; Peris et al., 2017; Chen et al., 2021) is a promising solution that can guarantee high-quality, error-free translations. It involves an iterative collaboration process between humans and machines, with multiple interactive steps to obtain a satisfactory translation.

Traditional IMT systems use a left-to-right completion paradigm (Barrachina et al., 2009; Knowles and Koehn, 2016; Wuebker et al., 2016) where human translators revise words in the translation prefix. This paradigm can be easily implemented using a prefix-constrained decoding strategy. However, this strict left-to-right manner limits its flexibility, as some translators may prefer to revise words in a different order. Recently, an alternative IMT paradigm has been proposed, allowing human translators to revise words at arbitrary positions in the translation. The essential technique for this paradigm is lexical-constrained translation (Cheng et al., 2016; Hokamp and Liu, 2017; Post and Vilar, 2018; Chen et al., 2020; Xiao et al., 2022), which leverages modified words as constraints to generate a satisfactory translation.

Although various IMT techniques have been proposed, there is currently a lack of a unified platform to fairly and systematically evaluate these methods, especially in an *end-to-end* manner. One of the difficulties is that previous IMT methods differ in their interactive policies, evaluation environments and metrics, making it challenging to fairly compare their performance and efficiency. Moreover, the evaluation methods previously employed, such as randomly deleting words or phrases and then comparing BLEU scores after one or several interactions (Weng et al., 2019; Chen et al., 2020; Xiao et al., 2022), are far from the real-world interactive experience. The final hypothesis may be not the golden translation and post-editing is still indispensable. Instead, the end-to-end paradigm, which evaluates the performance of IMT systems when the human translator finishes editing the translation, is a more accurate measure of the total cost of the whole iterative collaboration process.

In this paper, we introduce an open-source end-

---

*Work was done during internship at Tencent AI Lab.
†Corresponding author

to-end IMT system platform, namely IMTLAB, to fill this gap. This platform enables both academia and industry to quickly build IMT systems using state-of-the-art models, perform end-to-end evaluations, and diagnose weaknesses in IMT systems. IMTLAB treats the entire interactive translation process as a task-oriented dialogue, which includes a human-in-the-loop setting and allows IMT systems to leverage explicit human interventions to produce high-quality, error-free translations. Specifically, the user's goal is to obtain a correct translation, while the IMT system constantly generates translation based on the user's behaviors. During this process, the user iteratively performs editing operations to request the response of IMT systems until the pre-defined goal is achieved or the user loses patience with this interactive process. To support the flexible architectures of various IMT systems, we propose a general communication interface between IMT systems and users, where users are limited to five common types of editing operations: *keep*, *insert*, *replace*, *delete* and *blank-filling*. Then IMTLAB leverages the revised translation and the corresponding character-level editing operations from users to form lexical constraints for IMT systems. Moreover, we build a simulated or real interactive environment for IMT systems and introduce new evaluation metrics to better verify the effectiveness of IMT systems in an end-to-end manner.

We conduct simulated and manual experiments to systematically evaluate several popular IMT systems with this platform. Experimental results indicate that the prefix-constrained decoding approach still obtains the lowest editing cost in the end-to-end evaluation, while BiTIIMT (Xiao et al., 2022) achieves comparable editing cost with a better interactive experience, i.e., better success rate, lower average turns and response time. IMTLAB is also compatible with large language models, such as ChatGPT. Additional experiments show that ChatGPT yields a promising performance of editing cost in end-to-end evaluation, but it is not very robust to flexible lexical constraints. In summary, our contributions are as follows:

- To the best of our knowledge, we develop the first open-source end-to-end IMT system platform.[1]

- We design a general communication interface to support the flexible architectures of IMT systems,

[1]Codes are available at https://github.com/xuuHuang/IMTLab.

and construct a simulated or real interactive environment to achieve an end-to-end evaluation.

- We conduct simulated and manual experiments to systematically compare several popular IMT systems. The user interaction data collected during these experiments will be released to the community for future research.

## 2 Related Work

IMT has been developed to assist professional translators since the era of statistical machine translation (SMT) (Langlais et al., 2000; Foster et al., 1997; Barrachina et al., 2009; Cheng et al., 2016; Simianer et al., 2016). Recently, with the development of NMT (Bahdanau et al., 2015; Vaswani et al., 2017; Hassan et al., 2018), the field of IMT has undergone a major shift towards powerful deep neural models (Hokamp and Liu, 2017; Post and Vilar, 2018).

Early IMT systems typically adopt a left-to-right sentence completing framework (Barrachina et al., 2009; Knowles and Koehn, 2016; Wuebker et al., 2016; Peris et al., 2017), where users process the translation from the beginning and revise the leftmost error. However, this left-to-right approach is inflexible and inefficient for modifying critical translation errors near the end of the sentence. As a result, many researchers have explored alternative IMT paradigms that allow human translators to revise critical translation errors at any position.

One such paradigm is the pick-revise framework proposed by Cheng et al. (2016) to improve efficiency. With respect to neural translation models, lexical-constrained decoding (LCD) methods are employed to satisfy arbitrary constraints, such as Grid Beam Search (GBS) (Hokamp and Liu, 2017) and Dynamic Beam Allocation (DBA) (Post and Vilar, 2018). Then Weng et al. (2019) design a bidirectional IMT framework on top of LCD method. While LCD methods suffer from slow inference speed, some methods treat the lexical constraints as part of the input and let vanilla NMT directly learn the decoding strategy. For example, LeCA (Chen et al., 2020) and BiTIIMT (Xiao et al., 2022) are two such methods that employ soft and hard lexical constraints, respectively, to improve translation quality and inference speed. Despite the numerous IMT techniques, there is currently no unified platform to evaluate these methods in a fair and systematic end-to-end manner. To fill this gap, we develop the first open-source platform for building,

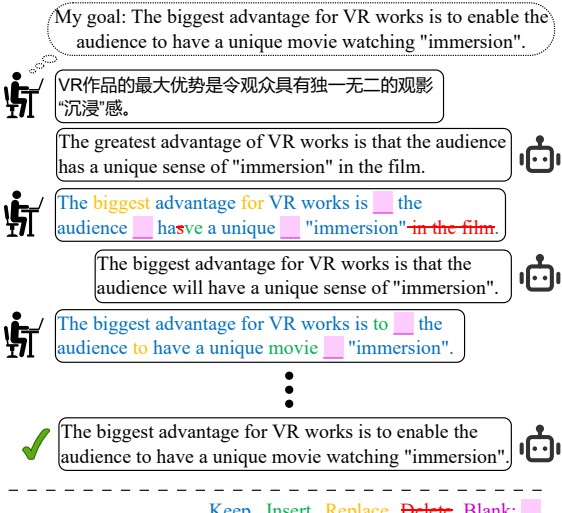

My goal: The biggest advantage for VR works is to enable the audience to have a unique movie watching "immersion".

VR作品的最大优势是令观众具有独一无二的观影"沉浸"感。

The greatest advantage of VR works is that the audience has a unique sense of "immersion" in the film.

The biggest advantage for VR works is ▮ the audience ▮ ha~s~ve a unique ▮ "immersion" ~in the film~.

The biggest advantage for VR works is that the audience will have a unique sense of "immersion".

The biggest advantage for VR works is to ▮ the audience to have a unique movie ▮ "immersion".

⋮

✓ The biggest advantage for VR works is to enable the audience to have a unique movie watching "immersion".

Keep  Insert  Replace  ~Delete~  Blank: ▮

Figure 1: Overview architecture of IMTLAB, involving the iterative collaboration process between a user and an IMT system. During this process, the user provides editing operations (i.e., *keep*, *insert*, *replace*, *delete* and *blank-filling*) interactively to query IMT systems until the user's goal is achieved or the user loses patience with this interactive process.

evaluating and diagnosing IMT systems.

Another research line related to this topic is online learning from translation memory or human feedback, which is particularly effective when translating documents in an unseen domain. This approach typically enhances the performance of NMT systems by fine-tuning through human-corrected translations (Turchi et al., 2017; Kothur et al., 2018; Peris and Casacuberta, 2019), or by utilizing TM-NMT frameworks (Bapna and Firat, 2019; Xia et al., 2019; Hao et al., 2023) or $k$NN-MT methods (Khandelwal et al., 2021; Zheng et al., 2021a,b; Meng et al., 2022; Wang et al., 2022; Dai et al., 2023; Du et al., 2023) to incorporate human-corrected translation pairs. In this paper, we focus on the interactive process within a single sentence, rather than the knowledge transfer across sentences. We leave the incorporation of online learning in our platform for achieving a stronger IMT system as a future direction.

## 3 IMTLAB

This section details the design of IMTLAB and its flexibility to support a wide range of experiments.

### 3.1 Overall Design

In this work, we consider the entire interactive translation process as a task-oriented dialogue with a human-in-the-loop setting (Li et al., 2017; Liu

et al., 2018; Zhang et al., 2019; Wang et al., 2021), since IMT involves an iterative collaboration process between humans and machines, with multiple interactive steps to obtain a satisfactory translation. As illustrated in Figure 1, the user's objective in this dialogue is to obtain a reference translation for the source sentence input. The IMT system is designed to produce translations based on the user's feedback, and the user provides editing operations (such as *keep*, *insert*, *replace*, *delete*, and *blank-filling*) to request the system's response. This interactive process continues until the user's objective is met or the user becomes frustrated with the system. In this way, IMTLAB allows the IMT system to leverage explicit human interventions to refine translations and enables researchers to evaluate different IMT systems in an end-to-end manner.

### 3.2 Communication Interface

In order to support the flexible architectures of various IMT systems, we design a general communication interface between IMT systems and users. Users are limited to five common types of editing operations: (*i*) *keep* words unchanged; (*ii*) *insert* a continuous word span at any position; (*iii*) *replace* a continuous word span with new one; (*iv*) *delete* a continuous word span at any position; (*v*) replace a continuous word span with a special placeholder (or just insert) to prompt the IMT system for *blank-filling*. These five common types could be combined to cover interactive policies in most IMT paradigms, including left-to-right completion, pick-revise framework, and arbitrary lexical constraints. To preserve user editing information, we maintain the record of editing operations at the character level, which is language-friendly. Therefore, user feedback can be decomposed into two parts, the revised translation and the corresponding operation tags, providing the IMT system with as much information as possible about the user's edits.

Figure 2 illustrates an example of how IMT-LAB records editing operations from different users. Specifically, when the IMT system produces an unsatisfactory translation that requires human intervention, users can perform the aforementioned editing operations according to their own goals. It's worth noting that the editing processes of different people may vary, such as Human 1 and 2. The revised translation and corresponding operation tags from user feedback are then used to query the IMT system for the desired translation. These different

**Source**: VR作品的最大优势是令观众具有独一无二的观影"沉浸"感。

**User Goal**: The biggest advantage for VR works is to enable the audience to have a unique movie watching "immersion".

**IMT System**: The greatest advantage of VR works is that the audience has a unique sense of "immersion" in the film.

- **Human 1 (prefix constraint)**

```
  Text: The biggest advantage for VR works is to that the audience has a
Operations: kkkk|--r--||----k----|rrr|-----k----|iii|-----------b-----------
            a unique sense of "immersion" in the film.
            -------------------b-------------------|
```

- **Human 2 (complex constraints)**

```
  Text: The biggest advantage for VR works is that the audience * have a
Operations: kkkk|--r--||----k----|rrr|-----k-----|bbbb|-----k-----|bbkkkdii|-
            unique sense of "immersion" in the film.
            --k---||---b---||----k-----||----d-----|k
```

Keep(k) Insert(i) Replace(r) Delete(d) Blank(b)

Figure 2: An example of the communication interface, where users perform editing operations (i.e., *keep*, *insert*, *replace*, *delete* and *blank-filling*) on the output of the IMT system according to their own goals. We show two different editing processes from users: one of which contains a prefix constraint, while the other one contains complex constraints. The editing operations are at the character level and the kept, inserted and replaced characters actually are lexical constraints. "*" is a special placeholder for the *blank-filling* operation.

editing operations bring different constraints, e.g., constraints at prefixes or arbitrary positions.

For the implementation details, when users perform *keep*, *insert* and *replace* operations, it is natural to directly label the revised translation with corresponding operations tags, i.e., "k", "i" and "r" tags. For the *deletion* operation, we preserve deleted text in revised translations and label them with "d" tags to explicitly display the *deletion* process, which may provide more editorial information to the IMT system. The *blank-filling* and *deletion* operations are very similar operations, with the difference that the former means the words are redundant, while the latter means new words are needed to replace original ones. Therefore, we use a similar way to record the editing process of the *blank-filling* operation, which labels deleted text in revised translations with "b" tags. When the user only wants to insert new words without deletion, we append the special placeholder "*" in the revised translation and label it with "b" tags. Additionally, when users perform the combined operations of *delete*+*insert* or *insert*+*delete* on a continuous character span, we treat these combined operations as *replace* operations in practice. Finally, all editing operations from users could be converted into a lexical-constrained template **t**, where the kept, inserted and replaced characters are lexical constraints for IMT systems.

Note that our current version only supports the keyboard-related operations mentioned above, but it is capable of evaluating most common IMT systems. We will leave the adaptation to other special operations like mouse clicks (Sanchis-Trilles et al., 2008; Navarro and Casacuberta, 2022) or screen touches for future work.

### 3.3 IMT System

During the interactive process, the IMT system must constantly generate translations based on the user's editing operations, which requires the system to support both normal and lexical-constrained translations. Specifically, given a source sentence **x**, the user first obtains an initial translation $\hat{\mathbf{y}}^{(0)}$ by querying the IMT system without any constraints:

$$\hat{\mathbf{y}}^{(0)} = \text{IMT}(\mathbf{x}) = \arg\max_{\mathbf{y}} P(\mathbf{y}|\mathbf{x}). \quad (1)$$

If this translation does not meet the user's requirements, the following interaction process begins. At the $i$-th turn, users modify the translation $\hat{\mathbf{y}}^{(i)}$ returned by the IMT system according to the user's goal $\mathbf{y}_O$, and corresponding editing operations are converted into a lexical-constrained template $\mathbf{t}^{(i)}$

to obtain the next translation $\hat{\mathbf{y}}^{(i+1)}$:

$$\mathbf{t}^{(i)} = \text{Policy}(\hat{\mathbf{y}}^{(i)}, \mathbf{y}_O).$$
$$\hat{\mathbf{y}}^{(i+1)} = \text{IMT}(\mathbf{x}, \mathbf{t}^{(i)}) \qquad (2)$$
$$= \arg\max_{\mathbf{y}} P(\mathbf{y}|\mathbf{x}, \mathbf{t}^{(i)}),$$

where constraints in $\mathbf{t}^{(i)}$ should appear in the translation $\hat{\mathbf{y}}^{(i+1)}$. This interactive process continues until we obtain $\mathbf{y}_O$. If the user loses patience with this interactive process, they could directly apply a post-editing strategy at any time to completely correct the translation.

### 3.4 Simulated Environment

We construct a simulated interactive environment where the IMT system interacts with user simulators instead of real users. Although there may be differences between simulated and real users, this setting enables researchers to quickly conduct a detailed analysis of IMT systems without any real-world cost. During the interactive translation process, the reference serves as an oracle to determine whether the words or phrases in the translation are correct, and the simulator provides a simulated user response on each turn according to their pre-defined interactive policy. The translation is only accepted when it matches the reference. In IMTLAB, we implement five common interactive policies for simulated users, including machine translation post-editing (MTPE), left-to-right sentence completion (L2r), random sentence completion (Rand), left-to-right infilling (L2rI) and random infilling (RandI):

- **MTPE:** The simulated user corrects the initial translation directly in one turn to match the reference using the optimal editing sequence based on the Levenshtein distance (Levenshtein, 1965).

- **L2r:** The simulated user first identifies the leftmost incorrect word and corrects it using a *replace* operation. Then a *blank-filling* operation is employed to remove the suffix. This interactive policy is widely used in traditional IMT systems (Barrachina et al., 2009).

- **Rand:** As advanced IMT methods (Hokamp and Liu, 2017; Post and Vilar, 2018) now allow users to modify errors at any position, we extend L2r policy to simulate this scenario. The simulated user randomly selects one correct word from the initial translation as the start point and uses *blank-filling* operations to remove the prefix and the suffix of it. In subsequent iterations, the user identifies the position of the last constraint in the new

translation, randomly selects a direction (left or right), and corrects the first incorrect word in that direction using a *replace* operation. Then *blank-filling* operations are used to remove the prefix and the suffix of this continuous word span.

- **L2rI:** Recently, some IMT systems (Peris et al., 2017; Xiao et al., 2022) employ a text-infilling policy that retains all correct words in a sentence and utilizes *blank-filling* operations to complete the remaining parts, so we develop L2rI policy to simulate this scenario. In this policy, the simulated user replaces all incorrect word spans with *blank-filling* operations and appends one correct word using an *insert* operation at the position of the leftmost incorrect word at each turn.

- **RandI:** This strategy behaves almost the same as L2rI, but it appends one correct word using an *insert* operation at a position near the preserved words. The position is randomly selected.

To model situations where users become impatient with the interactive process, we assume that the simulated user will lose patience if the constraints in L2r are not met. As Rand, L2rI, and RandI offer greater flexibility in selecting editing operations, we assume that the simulated user is more likely to tolerate constraint violations up to three times. If a constraint violation does occur, we will restart the interactive policies based on the current translation. In addition, the maximum interaction rounds threshold is determined by the number of editing sequences provided by MTPE to simulate situations where the interaction process is too inefficient for users to continue. If the number of interaction rounds exceeds this threshold, we assume that the simulated user will lose patience. MTPE is leveraged to correct the current translation once the simulated user loses patience with the interactive process.

### 3.5 Evaluation Metrics

The evaluation methods used in previous IMT methods, such as randomly deleting words or phrases and comparing BLEU scores after one or several interactions, do not accurately reflect real-world interactive experiences. To address this issue, IMTLAB introduces several end-to-end evaluation metrics for IMT systems that provide more accurate measures of the total cost of the entire iterative collaboration process.

- **Editing Cost (EC).** For the interactive experience, the editing cost is the most crucial metric,

rather than BLEU scores, as the user is expected to achieve their desired translation through the iterative collaboration process. To this end, we define the cost of each editing operation based on the actual number of required keystrokes:

- keep: 0
- insert: #chars inserted
- delete: 1
- replace: #chars inserted + 1
- blank-filling: 1

During the interactive translation process, we calculate the total editing cost by summing the cumulative cost of the user's editing operations at each turn. While we acknowledge that the cost of using a mouse or touchpad could be considered, we have chosen to ignore it in the current version for the sake of simplicity.

- **Success Rate (SR).** We track the rate of user satisfaction with the interactive process to implicitly reflect the interaction experience.

- **Consistency (Con.).** To measure the user's cognitive cost of new words or phrases, we calculate the average edit distance between two adjacent outputs from IMT systems.

- **Average Turns (AT).** We count the average turns of the entire interactive translation process for each source sentence.

- **Response Time (RT).** The average response time of IMT systems in the entire interactive process.

Among these metrics, the editing cost reflects the workload of users, while the other metrics reflect the user's experience with the interactive process.

## 4 Experiments

### 4.1 Setup

**Data.** We conduct simulated and manual experiments on English-German (En-De) and Chinese-English (Zh-En) language pairs in both directions. For En↔De , we use the WMT14 En-De dataset consisting of 4.5 million parallel sentences as the training set, newstest2013 as the development set, and newstest2014 as the test set. We preprocess the data similar to the script in fairseq except that we use sentencepiece (Kudo and Richardson, 2018) to tokenize the sentences and learn a joint vocabulary of 40k. For Zh↔En, the training data is from the WMT17 Zh-En dataset containing 20M parallel sentences. We also use sentencepiece to

preprocess the data and learn a joint vocabulary of 60k. To evaluate the IMT systems in both simulated and manual experiments, we randomly sample 500 or 100 sentence pairs from each original test set, respectively, for the end-to-end evaluation.

**Models.** We implement four popular IMT systems based on fairseq toolkit (Ott et al., 2019).

- **Prefix:** a vanilla Transformer model. This model only supports L2r, in which prefix-constrained decoding is used to produce translations.

- **DBA** (Post and Vilar, 2018): a vanilla Transformer model same as the one above. During decoding, it adopts the DBA method to satisfy the lexical constraints.

- **BiTIIMT** (Xiao et al., 2022): a Transformer model that learns with the bilingual text-infilling task and can fill missing segments in a revised translation.

- **LeCA** (Chen et al., 2020): a lexical constraint-aware Transformer model that simply packs constraints and source sentence together with a separating symbol to generate the final translation. We use the same augmented data as BiTIIMT. The pointer network is also used in this model.

All models use the Transformer-big architecture and share all embeddings. The learning rate is 7e-4 and the warmup step is 4000 for all models. The maximum update is set to 300k for vanilla Transformers and 400k for models using data augmentation. The batch size is set to 32k for the En↔De tasks and 64k for the Zh↔En tasks. For inference, we average 10 best checkpoints of each model and the beam size is 5. We run each model on a single NVIDIA Tesla V100 GPU with a batch size of 1.

### 4.2 Simulation Evaluation

We evaluate the end-to-end performance of IMT models using different interactive policies, including MTPE, L2r, Rand, L2rI and RandI[2]. In this simulation evaluation, we first record the editing cost of each model, where we average the editing cost of 500 source sentences. As shown in Table 1, MTPE serves as the baseline without interaction and all methods obtain similar editing cost using MTPE, meaning that the translation performance of them

---

[2]We run three experiments with different seeds for Rand and RandI interactive policies. The variance of editing cost in the random experiments can be found in Appendix A.

| Policy | En-De | | | | De-En | | | | Zh-En | | | | En-Zh | | | |
|---|---|---|---|---|---|---|---|---|---|---|---|---|---|---|---|---|
| | Prefix | DBA | BiTIIMT | LeCA | Prefix | DBA | BiTIIMT | LeCA | Prefix | DBA | BiTIIMT | LeCA | Prefix | DBA | BiTIIMT | LeCA |
| MTPE | 86.41 | 86.41 | 85.62 | 85.39 | 73.41 | 73.41 | 73.51 | 70.71 | 104.82 | 104.82 | 105.20 | 105.64 | 36.71 | **36.71** | 36.56 | 37.34 |
| L2r | **62.78** | 65.76 | **66.67** | 65.50 | **57.66** | 59.21 | **60.45** | 58.62 | **80.42** | 84.28 | 81.96 | **86.41** | 29.60 | 36.80 | 31.49 | **32.51** |
| Rand | / | 104.49 | 74.27 | 86.41 | / | 96.54 | 68.52 | 77.00 | / | 123.65 | 93.42 | 126.01 | / | 53.30 | 40.09 | 64.17 |
| L2rI | / | 70.54 | 67.80 | 68.77 | / | 65.17 | 62.45 | 63.27 | / | 87.92 | 84.79 | 90.92 | / | 43.29 | 42.42 | 47.87 |
| RandI | / | 80.18 | 68.08 | 70.52 | / | 72.15 | 63.00 | 64.23 | / | 98.80 | 82.87 | 88.84 | / | 50.28 | 43.05 | 47.65 |
| Avg. | 62.78 | 80.24 | 69.21 | 72.80 | 57.66 | 73.27 | 63.61 | 65.78 | 80.42 | 98.66 | 85.76 | 98.05 | 29.60 | 45.92 | 39.26 | 48.05 |

Table 1: The editing cost (↓) of each model with different interactive policies in simulation evaluation. The lowest cost in each column is in bold and "Avg." shows the average of the above four editing costs.

| | SR (↑) | | | | Con. (↓) | | | | AT (↓) | | | | RT (↓) | | | |
|---|---|---|---|---|---|---|---|---|---|---|---|---|---|---|---|---|
| | Prefix | DBA | BiTIIMT | LeCA | Prefix | DBA | BiTIIMT | LeCA | Prefix | DBA | BiTIIMT | LeCA | Prefix | DBA | BiTIIMT | LeCA |
| **En-De** | 98.2% | 88.1% | **98.6%** | 90.2% | **3.94** | 4.72 | 4.00 | 4.00 | 7.8 | 8.1 | 7.5 | **7.3** | 307 | 673 | **167** | 282 |
| **De-En** | 96.8% | 87.7% | **97.2%** | 89.6% | **3.50** | 4.45 | 3.64 | 3.54 | 8.2 | 8.4 | 7.9 | **7.6** | 293 | 654 | **158** | 273 |
| **Zh-En** | 97.6% | 87.1% | **98.2%** | 80.2% | 5.37 | 6.05 | **4.97** | 5.18 | 11.3 | 11.3 | 10.8 | **10.7** | 398 | 791 | **200** | 354 |
| **En-Zh** | **97.0%** | 85.5% | 96.3% | 55.2% | 4.41 | 6.22 | **4.32** | 5.41 | 11.2 | 11.2 | 11.0 | **10.7** | 325 | 759 | **170** | 310 |
| Avg. | 97.4% | 87.1% | **97.6%** | 78.8% | 4.31 | 5.36 | **4.23** | 4.53 | 9.62 | 9.74 | 9.28 | **9.05** | 331 | 719 | **174** | 305 |

Table 2: The success rate, consistency, average turns and response time (ms) of each method in simulation evaluation, where we average the results of each model on different interactive policies. "Avg." shows the average of the above four translation directions. The detailed results are shown in Appendix A.

| Model | Metric | En-De | De-En | Zh-En | En-Zh |
|---|---|---|---|---|---|
| BiTIIMT | BLEU | 54.55 | 54.31 | 46.37 | 49.55 |
| | CSR | 100% | 100% | 100% | 100% |
| LeCA | BLEU | 55.32 | 54.96 | 45.88 | 48.95 |
| | CSR | 99.55% | 99.30% | 98.51% | 98.21% |

Table 3: BLEU and Copy Success Rate (CSR) of BiTI-IMT and LeCA on the test sets with sampled constraints.

is very close due to the same training dataset. Current IMT methods achieve significant improvement over MTPE in most interactive policies, indicating the benefits of introducing an interactive process. When considering the same model with different interactive policies, L2r outperforms in most cases, showing that this conventional approach is still competitive. Surprisingly, Prefix achieves the best editing cost compared to other models in all translation directions. BiTIIMT achieves comparable performance with Prefix when using L2r and it is more robust to all interactive policies.

In addition to the editing cost, we also analyze the success rate, consistency, average turns and response time of these models, where we average the results of each model on L2r, Rand, L2rI and RandI. From Table 2, we observe that BiTIIMT obtains lower average turns than Prefix and has the best success rate, consistency and response time. These results demonstrate that BiTIIMT provides a better interactive experience than other methods while achieving comparable editing cost to Prefix.

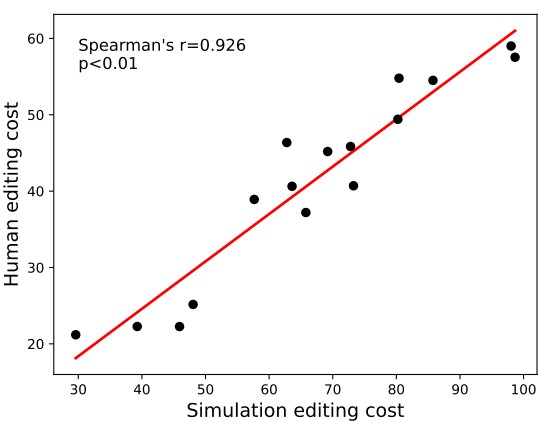

Figure 3: The correlation between the editing costs of simulated and real users.

We also note that lexical constraint-aware methods, such as LeCA, have the lowest success rate in end-to-end evaluation. As listed in Table 3, we randomly select some words or phrases as constraints to conduct experiments, following previous work (Chen et al., 2020), and find that the performance gap between BiTIIMT and LeCA is very small. These findings highlight the importance of end-to-end evaluation, which amplifies the error in each turn and closely resembles real-world interactive experience.

## 4.3 Human Evaluation

For human evaluation, real users interact with different IMT models to finish the translation task and we build the website for this purpose. To obtain

| | EC (↓) | | | | SR (↑) | | | | AT (↓) | | | | RT (↓) | | | |
|---|---|---|---|---|---|---|---|---|---|---|---|---|---|---|---|---|
| | Prefix | DBA | BiTIIMT | LeCA | Prefix | DBA | BiTIIMT | LeCA | Prefix | DBA | BiTIIMT | LeCA | Prefix | DBA | BiTIIMT | LeCA |
| **En-De** | 46.36 | 49.41 | **45.19** | 45.84 | 95.7% | 96.3% | **99.8%** | 96.3% | 4.6 | 3.9 | **3.7** | 4.1 | 264 | 617 | **154** | 247 |
| **De-En** | **38.91** | 40.07 | 40.63 | 37.20 | 98.7% | 98.7% | **99.7%** | 98.3% | 3.6 | **3.2** | 3.5 | 3.4 | 249 | 559 | **143** | 236 |
| **Zh-En** | 54.79 | 57.53 | **54.50** | 59.00 | **99.0%** | 95.3% | **99.0%** | 90.7% | 4.1 | **3.4** | 3.9 | 3.9 | 368 | 717 | **206** | 344 |
| **En-Zh** | **21.19** | 22.26 | 22.27 | 25.16 | **100%** | 99.0% | 99.7% | 84.3% | 3.7 | **2.8** | 3.1 | 3.0 | 296 | 632 | **176** | 284 |
| Avg. | **40.31** | 42.48 | 40.65 | 41.80 | 98.3% | 97.3% | **99.3%** | 92.4% | 4.0 | **3.3** | 3.6 | 3.6 | 294 | 631 | **170** | 278 |

Table 4: The editing cost, success rate, average turns and response time (ms) of each method in human evaluation, where we average the results of each model with three annotators. More results are shown in Appendix B.

user feedback on a real interactive experience, we provide a "MTPE" checkbox that the user could click to express their dissatisfaction with the interactive process. Therefore, in this experiment, the success rate is recorded as the unclicking rate of this checkbox, and we ignore the consistency metric, which is also reflected in user satisfaction. Specifically, we randomly sample 100 sentences from each test set and then ask three human translators to interact with IMT systems. Translators use flexible operations (such as *keep*, *insert*, *replace*, *delete*, and *blank-filling*) to interact with IMT models without any requirements.[3] The human evaluation results are listed in Table 4. Similar to the conclusion of the simulation evaluation, we observe that Prefix still obtains the lowest editing cost in the end-to-end human evaluation, while BiTIIMT achieves comparable editing cost to Prefix with a better interactive experience, i.e., better success rate, lower average turns and response time. We also calculate the Spearman's correlation between the average editing cost over four simulated policies and the average editing cost over three human translators, as shown in Figure 3. This result demonstrates a good correlation between simulated and manual experiments on IMTLAB. In addition, we observe that the editing cost of real users is significantly lower than that of simulated users. The real users feedback that *they learn to select the best operations by observing the output of IMT systems to improve their editing efficiency*, which currently could not be simulated by IMTLAB. We hope the release of real user interaction data will aid in the construction of such a simulated user.

## 4.4 Evaluation for ChatGPT

Large language models (LLMs) such as ChatGPT and GPT-4 have demonstrated remarkable machine translation ability during a chat. IMTLAB is compatible with these models by converting lexical

| | EC (↓) | | | | SR (↑) | | | |
|---|---|---|---|---|---|---|---|---|
| | **En-De** | **De-En** | **Zh-En** | **En-Zh** | **En-De** | **De-En** | **Zh-En** | **En-Zh** |
| MTPE | 81.58 | 66.56 | 100.33 | 35.39 | / | / | / | / |
| L2r | **74.00** | **57.38** | **89.68** | **33.95** | 63.0% | 71.0% | 60.4% | 53.0% |
| Rand | 126.55 | 114.20 | 158.56 | 59.34 | 28.4% | 25.2% | 25.8% | 17.4% |
| L2rI | 74.61 | 65.90 | 99.28 | 48.23 | **78.6%** | **80.8%** | **72.2%** | **67.8%** |
| RandI | 93.57 | 77.89 | 114.84 | 53.03 | 52.8% | 62.4% | 46.2% | 43.6% |
| Avg. | 92.18 | 78.84 | 115.59 | 48.64 | 55.7% | 59.9% | 51.2% | 45.5% |

Table 5: The editing cost and success rate of ChatGPT in simulation evaluation.

| | ChatGPT | | | | BiTIIMT | | | |
|---|---|---|---|---|---|---|---|---|
| | EC (↓) | SR (↑) | AT (↓) | RT (↓) | EC (↓) | SR (↑) | AT (↓) | RT (↓) |
| **En-De** | 48.03 | 90.0% | 3.2 | 3144 | 45.19 | 99.8% | 3.7 | 154 |
| **De-En** | 33.36 | 97.3% | 2.9 | 2627 | 40.63 | 99.7% | 3.5 | 143 |
| **Zh-En** | 54.85 | 91.3% | 3.3 | 3006 | 54.50 | 99.0% | 3.9 | 206 |
| **En-Zh** | 21.03 | 98.0% | 2.7 | 3455 | 22.27 | 99.7% | 3.1 | 176 |
| Avg. | 39.32 | 94.2% | 3.0 | 3058 | 40.65 | 99.3% | 3.6 | 170 |

Table 6: The editing cost, success rate, average turns and response time (ms) of ChatGPT and BiTIIMT in human evaluation.

constraints into natural language. We adopt the approach of BiTIIMT to use ChatGPT[4] as the IMT system that fills missing segments in a revised translation, and evaluate the performance using simulated and manual settings. More details for ChatGPT are presented in Appendix C.

Table 5 presents the editing cost and success rate of ChatGPT in simulation evaluation. It is evident that L2r is still in a leading position compared to other interactive policies. However, the editing cost of ChatGPT is worse than BiTIIMT on average, since the success rate of ChatGPT is very low, no more than $81\%$. ChatGPT struggles to satisfy lexical constraints, even though we explicitly require it to strictly follow the template.

We conduct the human evaluation for ChatGPT following the setting of the previous subsection and list all results in Table 6. Surprisingly, the editing cost of ChatGPT is better than BiTIIMT but the

---

[3]We inform translators that Prefix only supports the left-to-right completion manner.

[4]We call the gpt-3.5-turbo-0301 API.

success rate is still unsatisfactory. Additionally, we compare the editing cost of ChatGPT and BiTIIMT using `MTPE` in human evaluation. ChatGPT and BiTIIMT achieve 66.9 and 72 points, respectively, but this 5-point gap is reduced to 1.3 during the interactive process, indicating the unsatisfied success rate of ChatGPT hinders the editing cost.

## 5 Conclusion

In this paper, we introduce IMTLAB, an open-source platform for building, evaluating and diagnosing IMT systems. IMTLAB treats the whole interactive translation process as a task-oriented dialogue. To this end, we design a general communication interface to support the flexible architectures of IMT systems and a simulated or real interactive environment is further constructed for the end-to-end evaluation. Experiments demonstrate that the prefix-constrained decoding approach still achieves the lowest editing cost in the end-to-end evaluation, while BiTIIMT achieves comparable editing cost with a better interactive experience. IMTLAB is also compatible with LLMs, such as ChatGPT.

## 6 Limitations

In this section, we discuss the limitations and future research directions of our work:

- Although the simulated and manual experiments show a strong correlation, there is still a gap between simulated and real users. Real users could learn to select the best operations by observing the output of IMT systems, which can improve their editing efficiency. We hope that the release of real user interaction data will aid in the construction of such a simulated user in the future.

- To remove the effect of multiple translation references on the interactive process, human translators are required to strictly follow the same reference, rather than engaging in a more realistic, completely unconstrained manner. In the future, we would like to extend our human evaluation to a more complex setting.

- Our current evaluation is limited to four typical IMT systems and ChatGPT, excluding other IMT systems. In the future, we hope more researchers could construct other IMT systems in IMTLAB.

- In this work, we mainly focus on the interactive process within a single sentence, rather than knowledge transfer across sentences. We leave

the incorporation of online learning or translation memory in our platform for achieving a stronger IMT system as a future direction.

- In the current platform, words that are not modified or edited in any way are automatically considered as right words during the interactive process. However, there is an alternative interaction strategy where unmodified words are automatically considered incorrect. Actually, we could introduce a default system mode to unify the two interaction strategies, in which this system mode would trigger different automatic post-processing steps after user editing. In this way, the total cost of both the *keep* and *delete* operations is set to 1 and then one of these costs would not be calculated when using different system modes. We leave this refined framework as future work, providing the flexibility needed to accommodate more interaction strategies.

## Acknowledgements

We would like to thank the anonymous reviewers for their insightful comments. Shujian Huang is the corresponding author. This work is supported by National Science Foundation of China (No. 62176120, 62376116), the Liaoning Provincial Research Foundation for Basic Research (No. 2022-KF-26-02).

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

## A   Simulation Evaluation Results

We provide more detailed statistics of the simulation results in Table 7, including the success rate, consistency, average turns and response time of each method. The variance of editing cost in the random experiments is shown in Table 9.

## B   Human Evaluation Results

We record the evaluation results of three human translators, as shown in Table 8.

## C   ChatGPT

We design some prompts for machine translation with templates so that ChatGPT can complete the translation task with lexical constraints, thus can do interactive machine translation. We compare five different candidate prompts designed by human or advised by ChatGPT and choose the best one for simulated and manual experiments. We test five different prompts on the En-De test set using L2r interactive policy. The prompts and results are shown in Table 12. The temperature is set to 0 and max tokens is 200. For the initial translation, we just use the following prompt:

```
Translate the following [SRC] text to
[TGT]:[X]
```

where [SRC] and [TGT] are the source language and the target language, and [X] is the source sentence. For the translation task with lexical constraints, we adopt the following prompt:

```
Translate the [SRC] sentence by filling
in the [TGT] template. Strictly follow
the given [TGT] template and generate a
whole translation.
[SRC] sentence: [X]
[TGT] template: [T]
[TGT] translation:
```

where [T] is the lexical-constrained template string and "_" denotes a blank. More results of ChatGPT in simulation and human evaluation are shown in Table 10 and Table 11.

## D   Human interface

Figure 4 demonstrates the GUI of our platform. Figure 4(a) shows an initial translation of the given source sentence. Then the user can edit the translation in the target text area by inserting, deleting, etc. The newly inserted characters by the user are black.

The user can also use defined hot keys to replace a span with a blank placeholder or just insert a blank. The revised translation is shown in Figure 4(b), where the remaining texts in the area are lexical constraints. After clicking the "Translate" button, a new translation is generated by the backend IMT system, as shown in Figure 4(c). To distinguish between newly generated texts and lexical constraints, they have different colors. The user continues this cycle until the translation is satisfactory and clicks the "Submit" button to move on.

## E   Examples

Figure 5 shows two examples of human interactive translation processes. The font color in this figure is the same as the font color of the human interface. In the first example, the user revises the translation in a left-to-right manner, while the user in the second example adopts an infilling-style policy.

| | Policy | En-De | | | | De-En | | | | Zh-En | | | | En-Zh | | | |
|---|---|---|---|---|---|---|---|---|---|---|---|---|---|---|---|---|---|
| | | Prefix | DBA | BiTIIMT | LeCA | Prefix | DBA | BiTIIMT | LeCA | Prefix | DBA | BiTIIMT | LeCA | Prefix | DBA | BiTIIMT | LeCA |
| SR(↑) | L2r | 98.2% | 88.4% | 98.2% | 83.6% | 96.8% | 90.0% | 95.6% | 86.4% | 97.6% | 82.4% | 97.6% | 69.0% | 97.0% | 86.0% | 94.8% | 44.6% |
| | Rand | / | 63.8% | 96.3% | 86.9% | / | 60.9% | 93.3% | 83.1% | / | 65.8% | 95.3% | 75.7% | / | 58.6% | 90.2% | 46.3% |
| | L2rI | / | 100% | 100% | 93.2% | / | 100% | 100% | 93.0% | / | 100% | 100% | 83.2% | / | 99.8% | 100% | 52.8% |
| | RandI | / | 100% | 100% | 97.1% | / | 100% | 100% | 96.0% | / | 100% | 100% | 92.7% | / | 97.7% | 100% | 77.0% |
| Con.(↓) | L2r | 3.94 | 3.93 | 4.16 | 3.84 | 3.50 | 3.54 | 3.61 | 3.36 | 5.37 | 5.36 | 5.33 | 5.30 | 4.41 | 5.49 | 4.36 | 4.46 |
| | Rand | / | 5.85 | 4.14 | 4.25 | / | 5.86 | 3.71 | 3.91 | / | 6.37 | 5.08 | 5.87 | / | 8.28 | 4.39 | 5.85 |
| | L2rI | / | 4.76 | 3.89 | 3.92 | / | 4.29 | 3.75 | 3.33 | / | 6.78 | 4.84 | 4.79 | / | 5.78 | 4.34 | 5.68 |
| | RandI | / | 4.34 | 3.79 | 3.97 | / | 4.10 | 3.50 | 3.57 | / | 5.69 | 4.62 | 4.75 | / | 5.34 | 4.18 | 5.65 |
| AT(↓) | L2r | 7.8 | 7.2 | 8.2 | 7.0 | 8.2 | 7.7 | 8.6 | 7.6 | 11.3 | 9.4 | 11.8 | 9.7 | 11.2 | 10.0 | 12.1 | 8.9 |
| | Rand | / | 10.6 | 8.5 | 9.1 | / | 10.9 | 9.1 | 9.3 | / | 15.0 | 12.4 | 14.1 | / | 14.5 | 12.7 | 15.3 |
| | L2rI | / | 6.9 | 6.7 | 6.5 | / | 7.1 | 6.9 | 6.7 | / | 9.8 | 9.6 | 9.4 | / | 9.4 | 9.6 | 8.9 |
| | RandI | / | 7.8 | 6.5 | 6.6 | / | 7.8 | 6.9 | 6.7 | / | 11.0 | 9.3 | 9.4 | / | 10.7 | 9.6 | 9.7 |
| RT(↓) | L2r | 307 | 811 | 176 | 281 | 293 | 803 | 171 | 271 | 398 | 938 | 212 | 355 | 325 | 1072 | 177 | 294 |
| | Rand | / | 704 | 193 | 281 | / | 678 | 186 | 275 | / | 822 | 233 | 353 | / | 862 | 211 | 306 |
| | L2rI | / | 615 | 150 | 283 | / | 591 | 138 | 269 | / | 745 | 177 | 353 | / | 522 | 142 | 313 |
| | RandI | / | 563 | 149 | 284 | / | 542 | 137 | 276 | / | 659 | 176 | 356 | / | 581 | 151 | 326 |

Table 7: The success rate, consistency, average turns and response time (ms) of each method in simulation evaluation.

| | Metric | En-De | | | | De-En | | | | Zh-En | | | | En-Zh | | | |
|---|---|---|---|---|---|---|---|---|---|---|---|---|---|---|---|---|---|
| | | Prefix | DBA | BiTIIMT | LeCA | Prefix | DBA | BiTIIMT | LeCA | Prefix | DBA | BiTIIMT | LeCA | Prefix | DBA | BiTIIMT | LeCA |
| Human 1 | EC | 44.03 | 45.32 | 41.99 | 44.09 | 36.08 | 37.9 | 37.8 | 35.34 | 50.39 | 53.33 | 50.18 | 55.52 | 20.56 | 22.22 | 22.86 | 26.17 |
| | SR | 96% | 95% | 98% | 95% | 99% | 97% | 99% | 97% | 99% | 95% | 99% | 88% | 100% | 99% | 99% | 80% |
| | AT | 4.1 | 4.3 | 3.9 | 4.3 | 3.6 | 3.4 | 3.7 | 3.9 | 4.6 | 3.4 | 4.1 | 4.2 | 3.1 | 2.9 | 3.4 | 3.2 |
| | RT | 269 | 592 | 151 | 250 | 252 | 541 | 139 | 239 | 384 | 737 | 192 | 363 | 304 | 638 | 169 | 300 |
| Human 2 | EC | 49.48 | 50.21 | 45.87 | 46.81 | 39.92 | 42.20 | 41.24 | 36.77 | 57.40 | 58.50 | 53.30 | 58.14 | 21.44 | 22.14 | 22.95 | 26.88 |
| | SR | 95% | 95% | 99% | 97% | 97% | 100% | 100% | 99% | 100% | 95% | 100% | 93% | 100% | 99% | 100% | 84% |
| | AT | 4.2 | 2.8 | 3.5 | 4.7 | 2.8 | 2.4 | 3.4 | 3.4 | 3.3 | 3.0 | 3.7 | 3.7 | 3.0 | 2.2 | 2.7 | 3.3 |
| | RT | 262 | 513 | 139 | 245 | 245 | 462 | 131 | 241 | 366 | 581 | 205 | 342 | 296 | 447 | 169 | 286 |
| Human 3 | EC | 45.58 | 52.70 | 47.70 | 46.63 | 40.74 | 42.00 | 42.86 | 39.48 | 56.58 | 60.77 | 60.03 | 63.34 | 21.56 | 22.43 | 21.00 | 22.42 |
| | SR | 96% | 99% | 100% | 97% | 100% | 99% | 100% | 99% | 98% | 96% | 98% | 9100% | 100% | 99% | 100% | 89% |
| | AT | 5.4 | 4.5 | 3.8 | 3.3 | 4.0 | 3.8 | 3.5 | 2.9 | 4.4 | 3.7 | 3.9 | 3.9 | 5.1 | 3.4 | 3.2 | 2.4 |
| | RT | 264 | 617 | 154 | 247 | 249 | 559 | 143 | 236 | 368 | 717 | 206 | 344 | 296 | 632 | 176 | 284 |

Table 8: The editing cost, success rate, average turns and response time (ms) of each method in human evaluation.

| | En-De | | De-En | | Zh-En | | En-ZH | |
|---|---|---|---|---|---|---|---|---|
| | Rand | RandI | Rand | RandI | Rand | RandI | Rand | RandI |
| DBA | 0.161 | 0.592 | 2.784 | 0.048 | 3.702 | 0.197 | 0.120 | 0.024 |
| BiTIIMT | 0.678 | 0.403 | 0.001 | 0.151 | 0.217 | 0.318 | 0.020 | 0.015 |
| LeCA | 0.069 | 0.041 | 0.563 | 0.572 | 0.105 | 0.484 | 0.725 | 0.252 |

Table 9: The variance of editing cost in the random experiments.

| | Con. | | | | AT | | | |
|---|---|---|---|---|---|---|---|---|
| | En-De | De-En | Zh-En | En-Zh | En-De | De-En | Zh-En | En-Zh |
| L2r | 4.47 | 3.57 | 5.84 | 4.24 | 5.6 | 6.2 | 8.4 | 8.2 |
| Rand | 5.44 | 6.99 | 10.23 | 5.68 | 7.9 | 8.3 | 10.4 | 10.4 |
| L2rI | 4.30 | 3.69 | 6.22 | 4.25 | 6.1 | 6.2 | 8.6 | 8.5 |
| RandI | 3.61 | 3.21 | 5.79 | 3.77 | 6.1 | 6.3 | 7.6 | 7.9 |

Table 10: The consistency and average turns of Chat-GPT in simulation evaluation.

| | Metric | En-De | De-En | Zh-En | En-Zh |
|---|---|---|---|---|---|
| Human 1 | EC | 42.42 | 31.97 | 53.89 | 21.27 |
| | SR | 89% | 95% | 86% | 99% |
| | AT | 3.1 | 3.0 | 3.7 | 2.9 |
| | RT | 3548 | 3766 | 3426 | 4071 |
| Human 2 | EC | 51.26 | 35.70 | 56.55 | 22.62 |
| | SR | 90% | 98% | 93% | 96% |
| | AT | 3.7 | 3.0 | 3.2 | 2.9 |
| | RT | 2895 | 2132 | 2938 | 3175 |
| Human 3 | EC | 50.40 | 32.41 | 54.10 | 19.19 |
| | SR | 91% | 99% | 95% | 99% |
| | AT | 2.9 | 2.7 | 2.9 | 2.2 |
| | RT | 2989 | 1984 | 2654 | 3120 |

Table 11: The editing cost, success rate, average turns and response time (ms) of ChatGPT in human evaluation.

| Prompt | | Editing Cost | Success Rate |
|---|---|---|---|
| Translate the [SRC] sentence by filling in the [TGT] template. Strictly follow the given [TGT] template and generate a whole translation
[SRC] sentence: [X]
[TGT] template: [T]
[TGT] translation: | | 74.00 | 0.63 |
| Strictly follow the provided [TGT] template and information to generate a grammatically correct [TGT] sentence that accurately conveys the same meaning as the given [SRC] sentence. You must generate a complete sentence and any deviation from the template should be avoided.
[SRC] sentence: [X]
[TGT] template: [T]
[TGT] sentence: | | 99.88 | 0.22 |
| Use the provided [TGT] template and information to generate a sentence in [TGT] that conveys the same meaning as the given [SRC] sentence. Ensure that the sentence follows the given template exactly.
[SRC] sentence: [X]
[TGT] template: [T]
Complete [TGT] sentence: | | 95.24 | 0.42 |
| [SRC] sentence: [X]
[TGT] template: [T]
Create a [TGT] sentence using the given template and information that accurately translates the provided [SRC] sentence. You must conform to the template and generate the whole translation.
[TGT] sentence: | | 74.32 | 0.48 |
| [SRC] sentence: [X]
[TGT] template: [T]
Your task is to provide a German translation of the given English sentence. You must use the given [TGT] template and information exactly as provided without making any changes, and generate a complete translation.
[TGT] translation: | | 74.33 | 0.45 |

Table 12: Results of different prompts of ChatGPT.

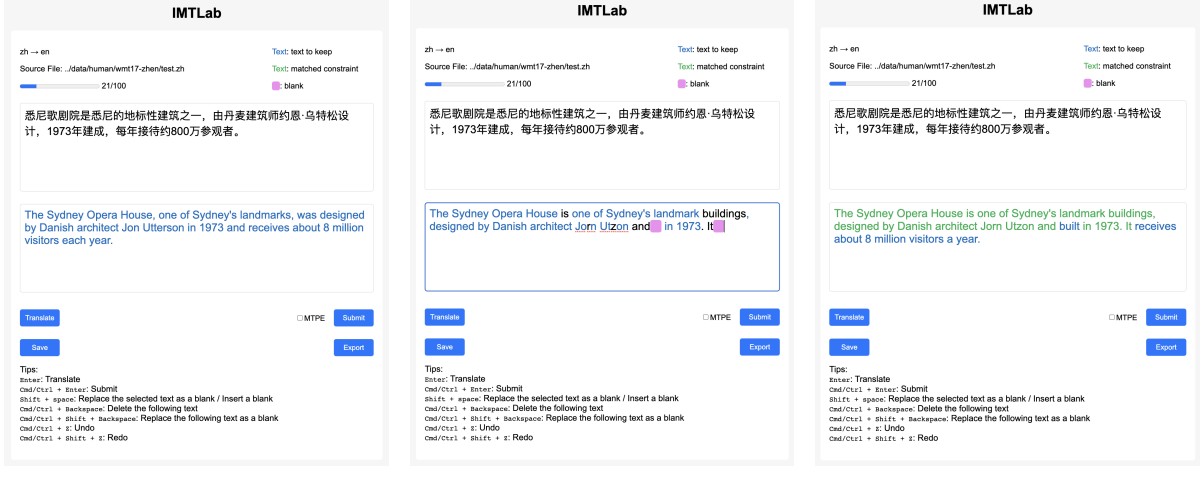

(a) The initial translation   (b) The revised translation   (c) The new translation

Figure 4: Demonstrations of the human interface.

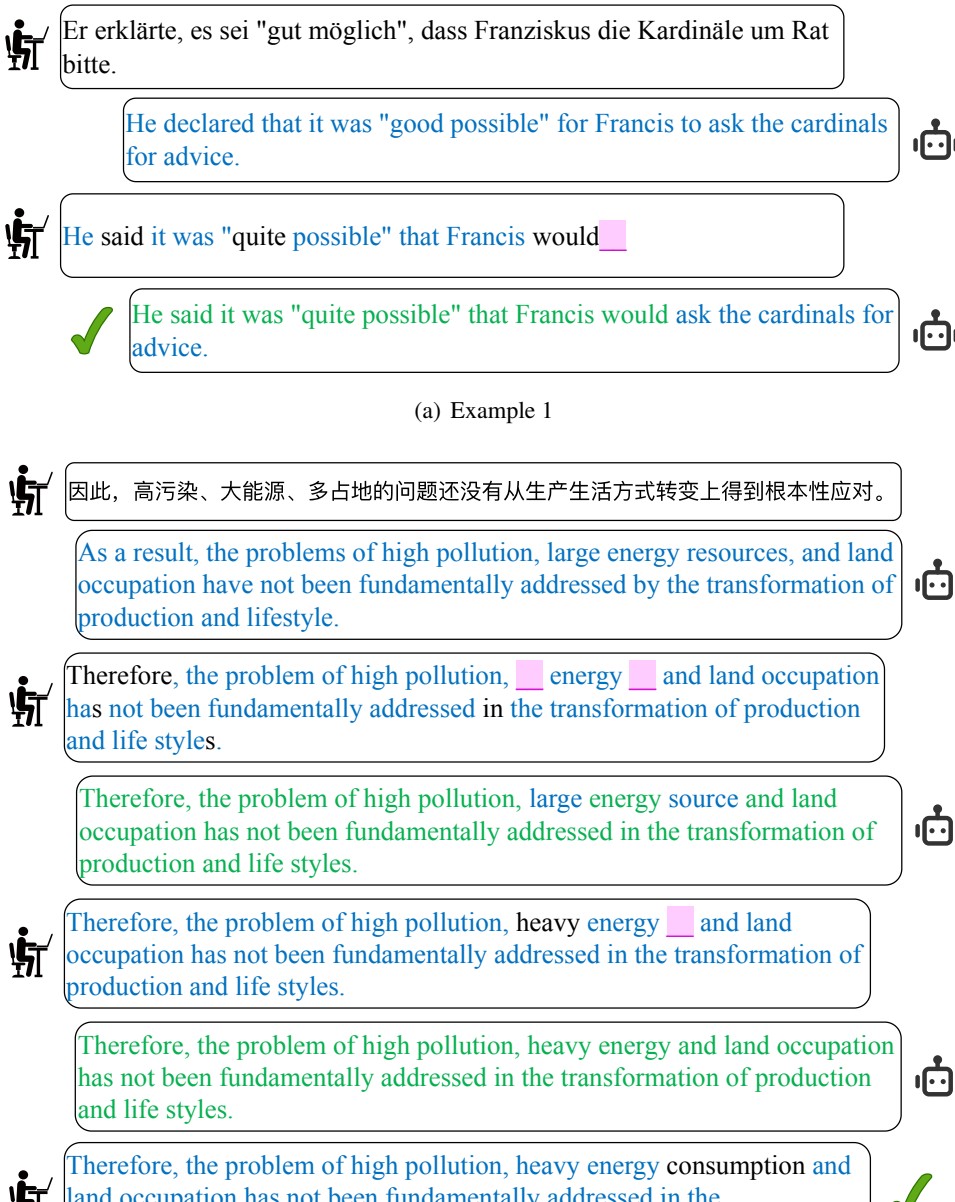

(a) Example 1

(b) Example 2

Figure 5: Examples of interactive translation processes