# OpenReview forum: "IMTLab: An Open-Source Platform for Building, Evaluating, and Diagnosing Interactive Machine Translation Systems"
_EMNLP/2023/Conference — EMNLP 2023 Main_

### Official Review · Reviewer_UJug · 2023-07-21

**Soundness:** 4

**Excitement:**

4: Strong: This paper deepens the understanding of some phenomenon or lowers the barriers to an existing research direction.

**Missing References:**

- "A Post-editing Interface for Immediate Adaptation in Statistical Machine Translation" presents a variant of interactive machine translation where phrases of a phrase-based MT system can be moved inidivudally.

**Paper Topic And Main Contributions:**

The authors present a unified toolkit for the evaluation if interactive machine translation approaches. Formally, they define a set of operations (keep, insert, replace, delete, blank filling) and policies (prefix decoding and several variations of simulating in-filling). Given a policy and a set of edits, the backing machine translation systems are queried to provide a step towards an improved translation. In an evaluation, four different models are compared employing the four different policies. Metrics include the model's response time and the average number of turns needed to arrive at the final translation. In simulated experiments the authors show that a traditional transformer model doing prefix-decoding performs best in most metrics, an in-filling approach comes however quite close.
Lastly, the authors present a short human evaluation, where first good correlation is shown to the simulated experiments.

**Questions For The Authors:**

- Line 323: I'm not entirely sure what MTPE is: Is the MT output corrected in 1 step? If not, which edit is applied first? In general it would be helpful to have examples for each policy.
- How is prefix decoding implemented in the policies and in the user interface? First removal of the remaining suffix and then blanking?
- It would be intersting to compare the overall editing time in the human evaluation.
- Line 450: What is the variance in the Rand experiments?
- Line 405: What is the success rate in the simulated experiments?

**Reasons To Accept:**

- The unified toolkit for evaluating interactive machine translation is likely to be very useful to compare interactive machine translation approaches.
- The proposed evaluation metrics and abstractions should further help in this regard.

**Reasons To Reject:**

- At times the paper is lacking important details which may hinder replicability.

**Reproducibility:**

3: Could reproduce the results with some difficulty. The settings of parameters are underspecified or subjectively determined; the training/evaluation data are not widely available.

**Reviewer Confidence:**

4: Quite sure. I tried to check the important points carefully. It's unlikely, though conceivable, that I missed something that should affect my ratings.

**Typos Grammar Style And Presentation Improvements:**

- The tables are somewhat too dense.

---

> ### Author Rebuttal · Authors · 2023-08-29
>
> Thank you for taking the time to provide a comprehensive review of our work. We appreciate your thoughtful feedback and constructive comments. Below, we address the concerns and questions you raised.
>
> > Line 323: I'm not entirely sure what MTPE is: Is the MT output corrected in 1 step? If not, which edit is applied first? In general it would be helpful to have examples for each policy.
>
> Yes, the output is corrected in 1 step. There are some examples for each policy in the supplementary materials. We will also include more examples for each policy in the appendix of the final version.
>
> > How is prefix decoding implemented in the policies and in the user interface? First removal of the remaining suffix and then blanking?
>
> In the **L2r** policy, the wrong suffix is directly replaced with a blank using the *blank-infilling* operation. In the human interface, the user can also use the hot key to directly replace texts to the right of the cursor with a blank.
>
> > It would be interesting to compare the overall editing time in the human evaluation.
>
> We apologize that the editing time is not recorded in the human evaluation. Thank you for pointing it out and we would like to incorporate this indicator in the future version.
>
> > Line 450: What is the variance in the Rand experiments?
>
> The variance of editing cost is shown in the following table. Most of the variance is less than 1. We will append these results in the future version.
>
> |         | ende  |       | deen  |       | zhen  |       | enzh  |       |
> |---------|-------|-------|-------|-------|-------|-------|-------|-------|
> |         | Rand  | RandI | Rand  | RandI | Rand  | RandI | Rand  | RandI |
> | DBA     | 0.161 | 0.592 | 2.784 | 0.048 | 3.702 | 0.197 | 0.120 | 0.024 |
> | BiTIIMT | 0.678 | 0.403 | 0.001 | 0.151 | 0.217 | 0.318 | 0.020 | 0.015 |
> | LeCA    | 0.069 | 0.041 | 0.563 | 0.572 | 0.105 | 0.484 | 0.725 | 0.252 |
>
> > Line 405: What is the success rate in the simulated experiments?
>
> To model situations where users become impatient with the interactive process, we assume that the simulated user will lose patience once the prefix constraints in **L2r** are not met. As **Rand**, **L2rI**, and **RandI** offer greater flexibility in selecting editing operations, we assume that the simulated user is more likely to tolerate constraint violations up to three times. In addition, if the number of turns exceeds the threshold defined as the length of MTPE editing sequence, the system also fails (it means that the interactive process could not bring improvement in editing efficiency). More details can be found from line 361 to line 378 on paper.
>
> > At times the paper is lacking important details which may hinder replicability.
>
> As mentioned in the paper, we will release our code, models and data after the acceptance of this paper.
>
> > Missing references and presentation improvements
>
> We will add more related work and revise the presentation in our final version.

---

### Official Review · Reviewer_HeBz · 2023-08-01

**Soundness:** 4

**Excitement:**

4: Strong: This paper deepens the understanding of some phenomenon or lowers the barriers to an existing research direction.

**Paper Topic And Main Contributions:**

This work contributes an open-source platform to build and evaluate IMT systems. It also proposes some evaluation criteria and performs the evaluation of several IMT policies over four different models.

On the one hand, I don't think the proposed editing cost is equally fair for all the IMT policies. In fact, I believe this is the reason why the prefix-based policy always yields the lowest cost: it is reduced to the number of character in the word correction + 1, while other more complex policies need to deal with word deletions.

On the other hand, author are using different IMT systems which have not been trained to perform with a particular policy. This, as far as I seen, what they are doing is not assessing which IMT policy performs better (overall) but how each IMT systems performs for each policy.

Finally, the proposed protocol limits the user interaction to several keyboard-related operations, missing the opportunity to use other peripherals to reduce the editing cost even more (e.g., the use of the mouse has showed significant improvements in the literature (e.g., Sanchis et al., 2008; Navarro and Casacuberta, 2022).

References
Sanchis-Trilles, G.; Ortíz-Martínez, D.; Civera, J.; Casacuberta, F.; Vidal, E.; Hoang, H. Improving Interactive Machine Translation via Mouse Actions. In Proceedings of the 2008 Conference on Empirical Methods in Natural Language Processing, Association for Computational Linguistics, Honolulu, HI, USA, 25–27 October 2008; pp. 485–494
Navarro, Ángel, and Francisco Casacuberta. 2022. "On the Use of Mouse Actions at the Character Level" Information 13, no. 6: 294. https://doi.org/10.3390/info13060294

**Questions For The Authors:**

I am curios as why consider the "keep" cost as 0 and the "delete" as 1. On the one hand, I don't think that a fixed cost of 1 is fair (even if it can be performed by keeping the "delete" key pressed, it is still not comparable with the cost of single-pressing any other key).

On the other hand, wouldn't it be better to leave the delete cost as 0 (i.e., not performing any operation to a word sequence equal to a deletion) and considering the cost of the word sequences to keep? On average, I believe there are going to be more word sequences to delete than to keep (considering that once you select a word sequence as "keep", this could be stored and automatically performed for the subsequent interactions). Of course, which cost to assign to that operation would be critical, but it could be leveraged by adding the use of the mouse (e.g, Peris et al. consider one mouse action for keeping single words and two for any other word sequence).

In any case, with this ramble I wanted to point out the importance of designing a more-complex, good evaluation scenario that could cover as many IMT protocols as possible.

**Reasons To Accept:**

It deploys an open-source platform which could lead towards a unification of designing and evaluating IMT protocols.

**Reasons To Reject:**

* The edit operations need to be carefully revised and extended to offer more freedom to IMT protocols.
* While interesting, the evaluation does not meet its intended purpose.

**Reproducibility:**

4: Could mostly reproduce the results, but there may be some variation because of sample variance or minor variations in their interpretation of the protocol or method.

**Reviewer Confidence:**

5: Positive that my evaluation is correct. I read the paper very carefully and I am very familiar with related work.

---

> ### Author Rebuttal · Authors · 2023-08-29
>
> Thank you for taking the time to provide a comprehensive review of our work. We appreciate your thoughtful feedback and constructive comments. Below, we address the concerns and questions you raised.
>
> > On the one hand, I don't think the proposed editing cost is equally fair for all the IMT policies.
>
> > I am curios as why consider the "keep" cost as 0 and the "delete" as 1. On the one hand, I don't think that a fixed cost of 1 is fair (even if it can be performed by keeping the "delete" key pressed, it is still not comparable with the cost of single-pressing any other key).
>
> > On the other hand, wouldn't it be better to leave the delete cost as 0 (i.e., not performing any operation to a word sequence equal to a deletion) and considering the cost of the word sequences to keep?
>
> Thank you for providing your valuable feedback on the new interactive policy and editing cost. We greatly appreciate your thoughtful suggestions, as they will help us enhance our platform to support a wider range of interactive policies and IMT systems.
> (a) In response to your concerns regarding the choice of considering the "keep" cost as 0 and the "delete" cost as 1, we understand the need for clarification and would like to provide a detailed explanation to address any potential confusion. In our platform, words that are not modified or edited in any way are automatically considered as **right** words during the interactive process. This approach aligns with the post editing behavior of users. Therefore, the cost associated with the "keep" operation is set to 0, as there is no additional effort or modification required for these words. Regarding the "delete" operation, our platform allows users to select a word span using a mouse or touchpad and then simply press the "delete" key. This method closely resembles the observed user behavior in human evaluation. While we acknowledge that the cost of using a mouse or touchpad could be considered, we have chosen to ignore it in the current version for the sake of simplicity. As a result, the total cost for the "delete" operation is set to 1. We hope this explanation clarifies the rationale behind our chosen costs for the "keep" and "delete" operations in our platform.
>
> (b) Your suggested policy has motivated us to develop a refined framework that can effectively accommodate both our proposed interaction strategy and your suggested approach. We appreciate the different perspective you have brought to the table. In your suggested setting, words that are not modified or edited in any way are automatically considered as **wrong** words during the interactive process. This implies that we would need to introduce a "new" key to facilitate the "keep" operation, similar to the "delete" operation mentioned earlier. Consequently, the cost associated with the "keep" operation would be set to 1.  To address these differing approaches, we could introduce a default system mode that unifies the two methods. This system mode would trigger different automatic post-processing steps after user editing. Under this unified approach, we would set the total cost of both the "keep" and "delete" operations to 1. Then, when using different system modes, one of these costs would not be calculated. It's important to note that our current communication interface is also compatible with this setting. We believe that this refined framework will provide the flexibility needed to accommodate both our proposed interaction strategy and your suggested approach.
>
> > On the other hand, author are using different IMT systems which have not been trained to perform with a particular policy. This, as far as I seen, what they are doing is not assessing which IMT policy performs better (overall) but how each IMT systems performs for each policy.
>
> Yes, we also want to emphesize that assesssing how each IMT system performs for each policy can bring out a deeper understanding of the relationship between the IMT system and each policy, which is helpful to guide the human translators on how to better interact with the IMT system.
>
> > Finally, the proposed protocol limits the user interaction to several keyboard-related operations, missing the opportunity to use other peripherals to reduce the editing cost even more
>
> We will append more discussions and cite these papers in our final version. We admit that our protocol is limited to some given operations at this time, yet it is capable of evaluating most common IMT systems. More efforts are needed to update our this protocal to support more operations, and we also hope that the community will participate in this project.
>
> > In any case, with this ramble I wanted to point out the importance of designing a more-complex, good evaluation scenario that could cover as many IMT protocols as possible.
>
> We agree with the importance of designing a more complex, good evaluation scenario that could cover as many IMT protocols as possible, which is also our goal. Our current version is able to cover the most of common evaluation scenarios. We will also be dedicated to this task and strive to make necessary progress. We hope that the community will participate in this project and collectively advance the field of IMT.

---

### Official Review · Reviewer_nTMj · 2023-08-07

**Soundness:** 4

**Excitement:**

4: Strong: This paper deepens the understanding of some phenomenon or lowers the barriers to an existing research direction.

**Paper Topic And Main Contributions:**

The paper describes an Open-Source Platform for evaluating interactive machine translation systems. For this, the platform offers an API that covers current IMT approaches and allows to simulate user behaviour. With the help of the platform, the papers evaluates several popular user strategies in combination with several different machine translation strategies in IMT, for several language directions. It also evaluates ChatGPT as (constrained) MT system. The paper uses various metrics for evaluation. Several of these are taken from the area of dialog modeling, as the paper treats IMT as a human-machine-dialog problem. Finally the paper validates the simulation on the metric of editing costs by comparing an evaluation with simulated users against an evaluation conducted with human users.

**Questions For The Authors:**

Question A: Given the current staten of user simulation and given the results from your paper, which advise would you give users of your platform on which simulation policy to use how, in order to predict human performance with an IMT system evaluated by this platform?

**Reasons To Accept:**

The paper offers a comprehensive platform for fast, automatic evaluation if IMT. The paper presents a comprehensive set of evaluations on different metrics with different systems and compares their results against results obtained with human translators.

**Reasons To Reject:**

The simulation of users still has room for improvement (as also commented by the authors) and will probably improve over the future work.

**Reproducibility:**

4: Could mostly reproduce the results, but there may be some variation because of sample variance or minor variations in their interpretation of the protocol or method.

**Reviewer Confidence:**

2: Willing to defend my evaluation, but it is fairly likely that I missed some details, didn't understand some central points, or can't be sure about the novelty of the work.

**Typos Grammar Style And Presentation Improvements:**

For Table 1: Maybe readability can be improved, if you would underline in every row the result for the best MT system.

---

> ### Author Rebuttal · Authors · 2023-08-29
>
> Thank you for taking the time to provide a comprehensive review of our work. We appreciate your thoughtful feedback and constructive comments. Below, we address the concerns and questions you raised.
>
> > Question A: Given the current staten of user simulation and given the results from your paper, which advise would you give users of your platform on which simulation policy to use how, in order to predict human performance with an IMT system evaluated by this platform?
>
> As mentioned in section 4.3, we have observed a positive correlation between the average editing cost across four simulated policies and the average editing cost across three human translators (see figure 3). Based on this finding, we recommend users to evaluate an IMT system using all supported simulated policies and take the average performance as the prediction of human performance. Furthermore, we encourage users to incorporate more simulated policies into our platform to capture the editing behavior of real translators as comprehensively as possible.

---

### Meta-Review · Area_Chair_F7V2 · 2023-09-23

**Recommendation:** 4

**Metareview:**

The reviewers were unanimous in their thoughts about this paper as being exciting and has potential to be a strong addition to the program.

---

### Decision · Program_Chairs · 2023-10-07

**Decision:**

Accept-Main

**Comment:**

The reviewers were unanimous in their thoughts about this paper as being exciting and has potential to be a strong addition to the program.